# Preliminary Estimations of Mars Atmospheric and Ionospheric Profiles from Tianwen-1 Radio Occultation One-Way, Two-Way, and Three-Way Observations

Min Liu [1], Lue Chen [2,3,*], Nianchuan Jian [4], Peng Guo [4], Jing Kong [2], Mei Wang [2], Qianqian Han [2], Jinsong Ping [3] and Mengjie Wu [4]

1 Shanghai Meteorological Service, Shanghai 200030, China; liu_min@shmh.gov.cn
2 Beijing Aerospace Control Center, Beijing 100094, China; kongjing@bacc.org.cn (J.K.)
3 National Astronomical Observatory, Chinese Academy of Sciences, Beijing 100101, China; jsping@bao.ac.cn
4 Shanghai Astronomical Observatory, Chinese Academy of Sciences, Shanghai 200030, China; jnnccc@shao.ac.cn (N.J.); gp@shao.ac.cn (P.G.); mjwu@shao.ac.cn (M.W.)
* Correspondence: chenlue@bao.ac.cn

**Abstract:** The radio occultation method, one of the methods used to provide planetary atmospheric profiles with high vertical resolution, was applied to China's first Mars mission, Tianwen-1. We carried out observations based on the Chinese Deep Space Network, and one-way, two-way, and three-way modes were used for Doppler observations from the Tianwen-1 spacecraft. We successfully obtained effective observations from Tianwen-1 on 22 and 25 March 2022. An inversion system developed for Tianwen-1 radio occultation observations enabled the derivation of neutral atmospheric density, pressure, temperature, and electron density profiles of Mars. Utilizing one-way tracking data, Martian ionospheric electron density profiles were retrieved at latitudes between 68.7 and 70.7 degrees (N). However, the presence of strong random walk noise in one-way tracking data led to poor inversion results. Meanwhile, Martian ionospheric electron density and neutral atmosphere profiles were extracted from two-way and three-way tracking data at latitudes between 55.1 and 57.0 degrees (S) on 22 March and at latitudes between 62.8 and 63.4 degrees (S) on 25 March. Importantly, our inversion results from Tianwen-1 maintained consistency with results from the Mars Express and the Chapman theory (mainly in the M2 layer). Through two days' observation experiments, we established and verified the occultation solution system and prepared for the follow-up occultation plans.

**Keywords:** radio occultation; Mars; Tianwen-1; atmosphere; ionosphere; one-way; two-way; three-way observations

## 1. Introduction

The radio occultation technique, initially proposed by Von Eshleman of Stanford University in 1962, employs the changes in the phase or amplitude of radio signals between the spacecraft and the ground station antenna sensors to probe the neutral atmosphere and ionosphere of the planet and its satellites [1]. The Jet Propulsion Laboratory (JPL) exploited radio links between the ground station and the Mariner 3 and 4 spacecraft to probe the atmosphere and other properties of Mars [2,3]. A large number of occultation measurements have been performed on Mars by the Mars Global Surveyor (MGS) [4,5]. These measurements revealed atmospheric waves and $CO_2$ condensation at the north pole, along with estimates of winds and other meteorological phenomena. The Mars EXpress (MEX) probe was launched in 2003 by the European Space Agency (ESA), and its measurements almost cover the whole latitude of Mars through multiple orbital adjustments [6]. The Mars Reconnaissance Orbiter (MRO) was launched in 2005 by the National Aeronautics and Space Administration (NASA), and it performed radio occultation measurements generally once per day, which aimed to measure the climate and history of water on Mars [7]. The

MAVEN (Mars Atmosphere and Volatile Evolution Mission) in the US launched in 2013 and was the first spacecraft to have the main purpose of exploring the upper atmosphere of Mars [8]. In addition to the atmospheric measurements taken from occultation, gravitational parameters and geometric dimensions of the planets can also be determined by the measurements, such as sizes, shapes, and masses [9,10].

In addition to Mars, the radio occultation technique has also been used for other planets in the solar system, such as Venus, Jupiter, Saturn and its rings, Uranus, and Neptune and its moon Triton (excluding Earth) [11–17]. With the rise of the Global Navigation Satellite System (GNSS), the radio occultation technique has been widely employed for examining the Earth's atmosphere [18,19]. Researchers have also advanced scientific works, including on the processing of data from Mars radio occultation observations and the analysis of characteristics within the Mars atmosphere and ionosphere [20–24].

However, China lacked independent radio occultation observations of Mars until the Tianwen-1 spacecraft launched in 2020. Tianwen-1 entered the remote sensing orbit and began scientific exploration after communication with the Mars rover had been completed. Our group conducted an occultation observation from Tianwen-1 in August 2021, before the Sun transit outage [25], though its effectiveness was diminished by the solar winds and a limited observation time of around two minutes (typically 6–7 min). In order to enhance observations, we applied for observation with stations from the Chinese Deep Space Network (CDSN). CSDN consists of three Chinese Deep Space Stations (CDSS): Jiamusi (JM) station, Kashi (KS) station, and Argentina (AG) station. All the ground stations are equipped with high-precision water vapor microwave radiometers and GNSS dual-frequency receivers for Earth media correction [26]. This led to multiple Doppler observations for Tianwen-1 in March 2022, encompassing one-way, two-way, and three-way observation modes.

The Doppler residuals (Observed minus Computed) were rigorously evaluated for all three observation modes, and then the neutral atmospheric density, pressure, temperature (below 40 km), and ionospheric electron density (above 40 km) of Mars were reconstructed. A comparison of our results with data from the MEX team and the Chapman theory model demonstrated the reasonability and accuracy of our observations, and inversion outcomes were illuminated. This marks a step forward in China's contribution to understanding Martian atmospheric features through radio occultation techniques.

## 2. Three Observation Modes for Tianwen-1

Mars radio scientific research is supported by deep space exploration stations equipped with large antennas for receiving electromagnetic signals and Doppler data processing equipment. The 35 m deep space station in the Kashi region of northwest China (referred to as KS) and the 66 m deep space station in the Jiamusi region of northeastern China (referred to as JM) were used for our radio occultation observation experiments. Both of these ground stations constitute integral components of the CDSN.

According to the link relationship between the station and the spacecraft, tracking modes can be categorized into one-way, two-way, and three-way configurations (depicted in Figure 1). The one-way tracking mode is notably straightforward, solely comprising a downlink path. In this mode, the ultra-stable oscillator (USO) aboard the Tianwen-1 spacecraft serves as the signal source, and the downlink receiving station is on the ground. In contrast, both the two-way and three-way tracking modes incorporate an uplink path alongside the downlink path, and the signals of the up-down links are coherent. The difference between the two-way mode and the three-way mode is that the uplink and downlink stations of the two-way mode are the same station, while the uplink and downlink stations of the three-way mode are two different stations [27].

Within the two-way and three-way observation modes, the ground station's hydrogen atomic clock serves as an exceptionally stable time and frequency reference, and the Allen variance of its 100–1000 s integration is about $10^{-15}$, which can reach 0.1 mm/s velocity measurement accuracy [28]. Conversely, the one-way open-loop mode relies on the spacecraft's carried USO to govern the downlink signal. However, the typical Allen

variance of the onboard USO in the Tianwen-1 mission, while $10^{-11}$, cannot meet the velocity precision requirement of 1 mm/s (corresponding to 30 mHz for one-way tracking and 60 mHz for two/three-way tracking) for single-frequency inversion, so the inversion accuracy of the one-way open-loop mode will be limited (the details of this will be discussed in Section 4.2).

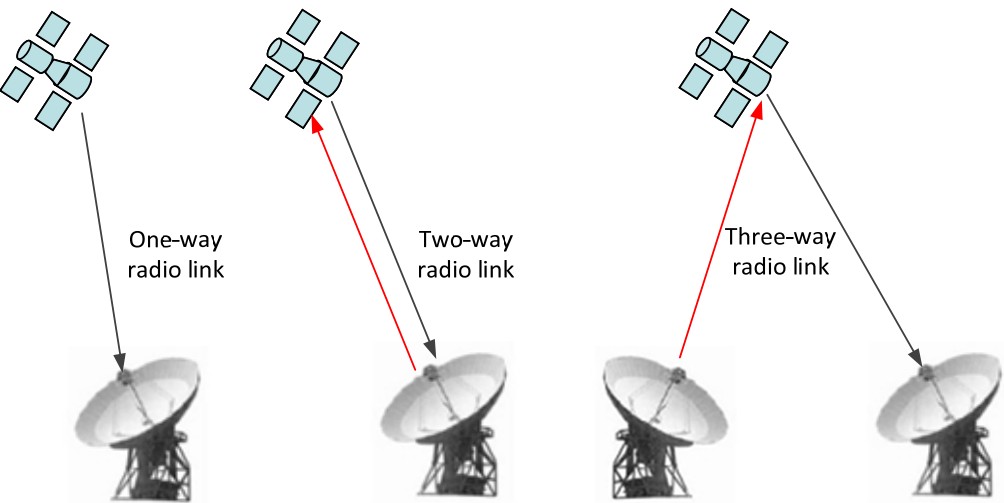

**Figure 1.** One-way, two-way, and three-way observation modes.

## 3. Radio Occultation Inversion Technique for Tianwen-1

This section is a summary of classical Radio Occultation theory with the assumption of a spherical planet.

The occultation coordinate system in Figure 2 is defined as follows:

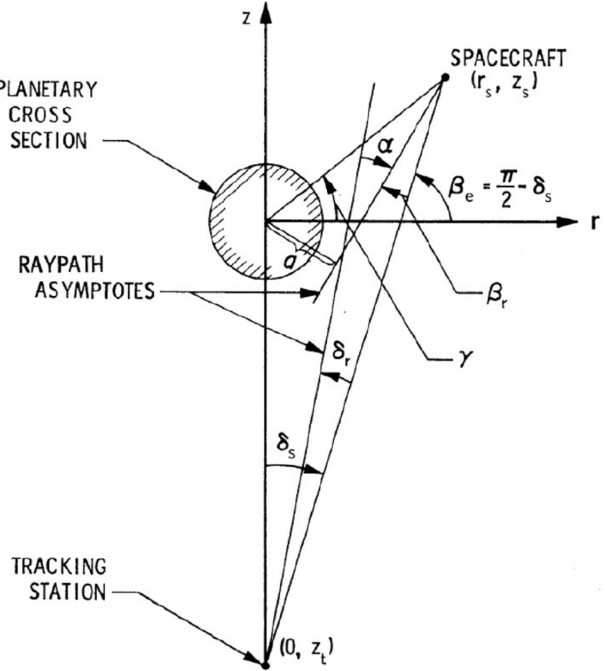

**Figure 2.** Geometry path for the radio occultation link between Earth and Mars [11].

Origin: The center of Mars.

Y-axis: the unit vector pointing from the observation station to the center of Mars.

X-Y plane: formed by the Y-axis vector and the vector connecting the observation station to the spacecraft.

X-axis: perpendicular to the Y-axis, lying within the X-Y plane.

$\delta_s$: The angle between the vector from the observation station to the spacecraft and the vector extending from the station to the center of Mars.

$\delta_r$: The angle between the vector from the observation station to the spacecraft and the vector representing the asymptotes of the ray path at the station's point.

$\beta_e$: The angle between the X-axis vector and the vector originating from the observation station to the spacecraft.

During the occultation period, the frequency of the electromagnetic wave received by the downlink station is different from the corresponding transmission frequency, and the difference is called the Doppler frequency shift. The Doppler frequency shift includes the Mars medium influence, the Earth medium influence, the interplanetary medium influence (when the observation direction is in proximity to the sun), the gravitational redshift, the gravitational delay, and the influence caused by orbital motion, etc. [2,29].

The geometry path for the radio occultation link between the Earth ground (tracking) station and Mars spacecraft Tianwen-1 is shown in Figure 2, and the meanings of each angle in Figure 2 are described in the appendix of Fjeldbo et al., 1971 [11].

The receiving and sending frequency of Tianwen-1 is a single frequency in the X band. The occultation observation method based on a single frequency is as follows:

$$\Delta f = \left( -f_s \frac{v_{rs}}{c} \sin \beta_r - f_s \frac{v_{zs}}{c} \cos \beta_r + f_s \frac{v_{zs}}{c} \right) \tag{1}$$

where $\delta_s = 0$, $\delta_r = 0$, $\beta_e = \pi/2$ in Figure 2, while ground stations on Earth are far away; $f_s$ is the transmission frequency at Tianwen-1, c is the speed of light in a vacuum, and, $v_{rs}$, $v_{zs}$ are the radial (relative to the center of mass of Mars) and z-direction velocities of the spacecraft, respectively.

With the assumption of spherical symmetry, according to the Bouguer formula, a bending angle sequence with impact parameters $a$ could be calculated from the upper to the lowest layer of Mars' atmosphere:

$$\begin{cases} \alpha(a) = \beta_r(a) \\ a = \sqrt{(r_s^2 + z_s^2)} \cos(\gamma + \beta_r) \end{cases} \tag{2}$$

where $a$ is the distance from the Mars center of mass to the ray path.

The refractivity index profile $n$ could be derived from the bending angle by following the Abelian integral:

$$n(r_1) = \exp \frac{1}{\pi} \int_{a_1}^{\infty} \frac{\alpha(a)}{\sqrt{a^2 - a_1^2}} da \tag{3}$$

where $a_1$ is the initial impact parameter corresponding to the radius at the tangent point $r_1$ for the ray of closest approach, and the refractivity could be calculated from $N(r) = (n(r) - 1) \times 10^6$.

In the ionosphere, the relationship between refractive index and electron density is [30,31]:

$$N(r) \approx \frac{\kappa_e}{f_0^2} N_e(r) \tag{4}$$

where $N_e$ is electron density, $\kappa_e \approx \frac{r_e c^2}{2\pi} \times 10^6$, and $r_e = 2.8179 \times 10^{-15}$ m is the classical Coulomb electron radius.

In the neutral atmosphere of Mars, the relationship between refractive index and molecular density is approximately written as:

$$N(r) \approx \sum_i f_i \kappa_i n_{n(r)} \tag{5}$$

where $\sum_i f_i \kappa_i = 1.804 \times 10^{29}$ m$^3$, $f_i$ is the ratio of different atmospheric components, and $n_n(r)$ is the atmospheric density of Mars.

In an ideal atmosphere, the Pressure profile is obtained according to the Hydrostatics Balance Equation:

$$\frac{\partial P(r)}{\partial r} = n_n(r)\overline{m}g_M \tag{6}$$

where $\overline{m} = 7.221 \times 10^{-26}$ kg is the average molecular mass of the Martian atmosphere and $g_M = 3.7$ m/s$^2$ is the Gravitational acceleration of the Martian surface, and the pressure $P(r)$ at tangent point $r$ could be obtained from Equation (6). Finally, the Temperature $T(r)$ could be inverted from the Ideal Gas Law.

## 4. Result and Discussion

### 4.1. Observation Data from Tianwen-1

On 22 and 25 March 2022, we conducted comprehensive observations encompassing both the ingress and egress occultation phases of Tianwen-1. These joint observations were conducted simultaneously at the Jiamusi and Kashi deep space stations, integral components of the CDSN, depicted as JM and KS, respectively, in Figure 3. Over the course of these observations, a total of ten distinct observation arcs were captured. Detailed information is outlined in Table 1.

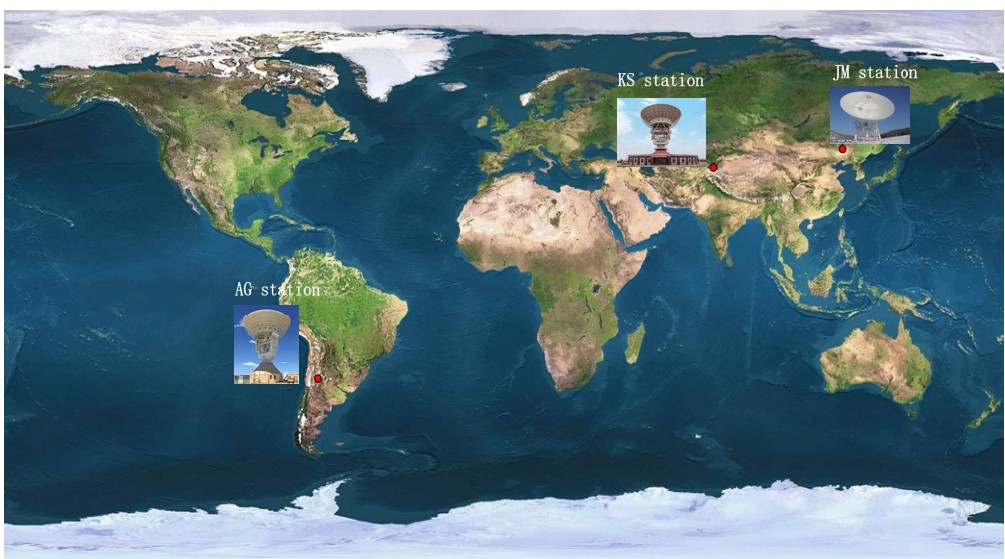

**Figure 3.** The location of CDSN.

**Table 1.** Observation information about Tianwen-1 and MEX.

| Arc No. | Time (UTC) | Up/Down | Occultation Type | Link Type | Band |
|---|---|---|---|---|---|
| arc-1 | 22 March, 02:25:01–02:35:38 | KS | ingress | 1-way | X |
| arc-2 | 22 March, 03:59:01–03:59:10 | | | | |
| arc-3 | 22 March, 03:59:01–03:59:10 | | | | |
| arc-4 | 22 March, 03:59:11–04:04:21 | KS/KS | egress | 2-way | X |
| arc-5 | 22 March, 03:59:11–04:04:21 | KS/JM | egress | 3-way | X |
| arc-6 | 25 March, 01:13:40–01:23:52 | JM | ingress | 1-way | X |
| arc-7 | 25 March, 02:52:12–02:52:19 | | | | |
| arc-8 | 25 March, 02:52:12–02:52:19 | | | | |
| arc-9 | 25 March, 02:52:20–02:55:00 | JM/JM | egress | 2-way | X |
| arc-10 | 25 March, 02:52:20–02:55:00 | JM/KS | egress | 3-way | X |
| MEX-0112 (2004) | 15 July, 15:23:00–15:35:23 | DSS-65/DSS-65 | ingress | 2-way | X |

DSS-65 refers to Deep Space Network Station 65, a facility operated by the United States of America and situated in Madrid, Spain. This station is outfitted with a 34-m radio telescope and played a key role in supporting the tracking mission of MEX.

Some observation arcs had shake-hand communication between the spacecraft and ground station, so the type was vacant (arc-2,3,7,8). MEX-0112 was observed from spacecraft MEX by station DSS-65 on 22 July 2004. The atmospheric and ionospheric profiles of MEX-0112 observation are discussed in Section 4, so the details of MEX-0112 are also outlined in Table 1. For successful tracking (arc-1,4,5,6,9,10, and MEX-0112), the related occultation information of latitude, longitude, and solar zenith angle (SZA) is outlined in Table 2.

**Table 2.** Occultation information is needed for successful tracking.

| Arc No. | Latitude (Deg.) | Longitude (Deg.) | SZA (Deg.) |
|---|---|---|---|
| arc-1 | 68.7–70.5 (N) | 204.8–219.2 | 74.6–76.8 |
| arc-4 | 55.1–57.0 (S) | 327.2–330.9 | 80.1–81.3 |
| arc-5 | 55.2–57.0 (S) | 327.3–330.9 | 80.1–81.3 |
| arc-6 | 70.2–70.7 (N) | 146.2–156.4 | 77.7–78.9 |
| arc-9 | 62.8–63.3 (S) | 295.2–297.1 | 88.4–88.8 |
| arc-10 | 62.8–63.4 (S) | 295.3–297.1 | 88.4–88.8 |
| MEX-0112 (2004) | 21.7–26.2 (N) | 243.3–240.5 | 77.9–78.0 |

4.1.1. Tianwen-1 Working Orbit

The orbital semi-major axis of Tianwen-1 is about 8000 km, the eccentricity is 0.59, the inclination is about 84 degrees, and the period is about 7 h. More information about orbit determination can be found in Table 3.

**Table 3.** Orbit determination strategy.

| Parameter | Source/Value |
|---|---|
| Coordinate System | MEME2000 |
| Central body | Mars |
| Mars Gravity Model | JGMRO120, 120 × 120 |
| N-body perturbation | Sun, major planets, and the Moon |
| Solar and Planetary GM | DE-436 |
| Solar radiation pressure model | Fixed area-mass ratio |
| Solar Radiation Coefficient, CR | The nominal value of the coefficient of SRP is 1.4 |
| Atmospheric correction model | MCD 5.2 |
| Relativity perturbation | Schwarzschild |
| Parameters to solve | Position and velocity, SRP coefficient, Systematic error of range data Empirical acceleration |
| Data weights | Range: 3 m Doppler: 1 mm/s VLBI delay: 0.3 ns VLBI delay rate: 0.3 ps/s |

The accuracy of orbit determination is also verified by orbit overlap, as shown in Figure 4.

The first sub-arc is from 21 March 2022 T00:00:00 to 24 March 2022 T08:00:00(UTC), the second sub-arc is from 24 March 2022 T00:00:00 to 27 March 2022 T08:00:00(UTC), and the overlap duration is 8 h. The position difference of overlap is about 65 m, and the speed difference is 3.2 cm/s. The overlap arc differences of other arcs are shown in Table 4. In general, the position accuracy of Tianwen-1 in this orbit is less than 70 m, and the velocity accuracy is about 3 cm/s.

The determination of Tianwen-1's orbit was supported by both Doppler and Very Long Baseline Interferometry (VLBI) tracking data captured at various CDSN stations. In Figure 5, the orbital Doppler residuals show the velocity precision of a common orbit determination for Tianwen-1 from Earth ground tracking station JM on 6 February 2021, which is about 0.1 mm/s [24]. Open-loop (red point) is the novel Doppler extractor, and baseband (blue point) is the classical Doppler extractor equipped on a ground station. It is shown in Figure 5 that the precision of the open-loop result is higher than that of the

baseband. The open-loop receiver was used for our experiment and is also mentioned in Section 4.1.2.

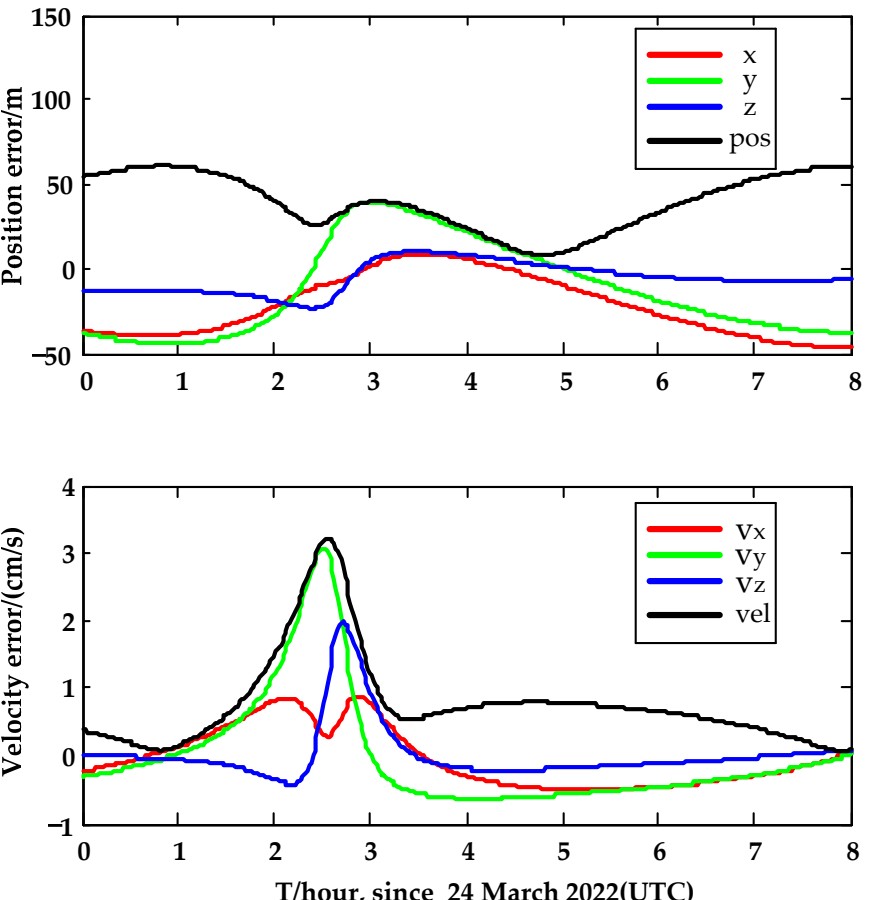

**Figure 4.** Orbit differences and uncertainty on position/velocity in the overlap arc.

**Table 4.** Results of precision orbit determination around occultations.

| Sub Arcs | Orbit Determination Arcs | | | | Overlap Arcs | | |
| | Start (MJD) | End (MJD) | Length (Day) | O−C (mm/s) | Length (Day) | Position Difference (m) | Velocity Difference (m/s) |
|---|---|---|---|---|---|---|---|
| 1 | 59,369.44 | 59,371.98 | 2.54 | 0.41 | - | - | - |
| 2 | 59,371.35 | 59,373.87 | 2.52 | 0.42 | 0.63 | 25.06 | 0.015 |
| 3 | 59,373.00 | 59,375.92 | 2.92 | 0.53 | 0.87 | 35.14 | 0.027 |
| 4 | 59,375.10 | 59,377.99 | 2.89 | 0.77 | 0.82 | 17.24 | 0.012 |

After entering working orbit, the orbit determination and occultation observations experienced a noticeable impact stemming from the solar transit outage commencing in August 2021. Due to the solar transit, our two Martian occultation observations failed in August 2021, so we embarked on a fresh series of occultation experiments in March 2022. For all the observation arcs in 2022 (listed in Table 1), the angles of Sun, Earth, and Prob (SEP) were around 50 degrees, which indicates a favorable condition where the Doppler observations remain largely impervious to solar interference.

Moreover, the observation angles between the signal path and the Tianwen-1 orbital plane were 6.1 degrees on 22 March and 3.7 degrees on 25 March 2022. A smaller observation angle signifies a more optimal and accurate occultation. The posture of the orbital plane toward Earth is called edge-on, while the angle is 0.

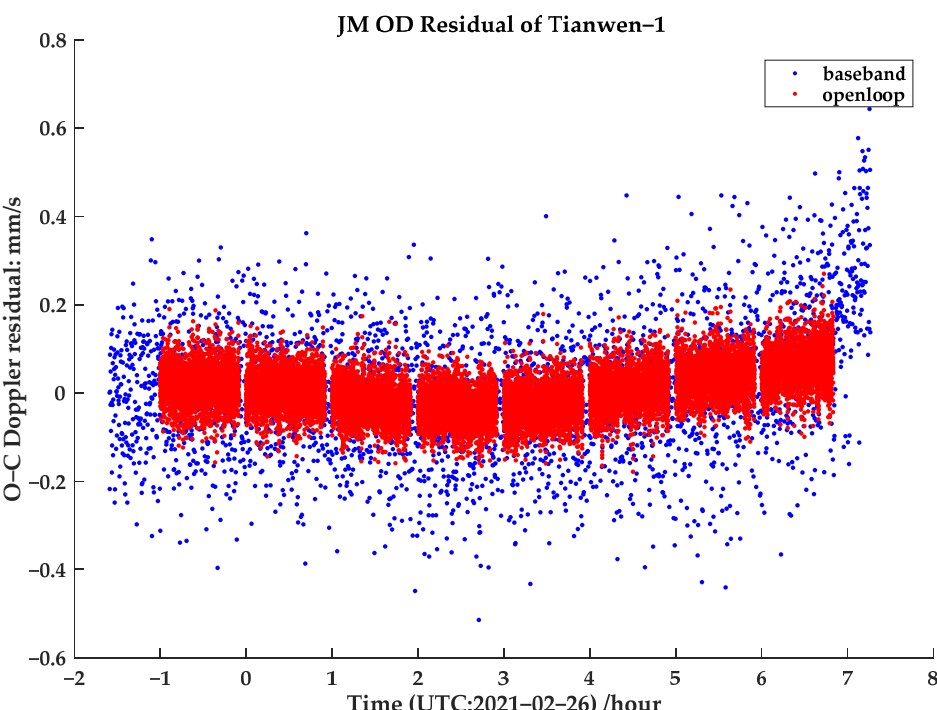

**Figure 5.** Orbital fitting residuals of Tianwen-1.

### 4.1.2. Zero-Level Data Processing

The fundamental dataset, referred to as zero-level data, comprises the initial recordings derived from the open-loop receiver situated on the station. Zero-level data processing provides Doppler observations for subsequent occultation inversion. A novel method based on local correlation was used in the zero-level data processing for our occultation experiment [25]. The signal processing procedure of this method was based on FFT (Fast Fourier Transformation), CZT (Chirp-Z Transform), and correlation-combined techniques. This method has been successfully utilized in Tianwen-1 and MEX high-precision orbit determination missions. In this Mars/Earth radio occultation experiment, the Tianwen-1 downlink carrier signal was processed by the open-loop measurement software mentioned in [25]. The Doppler sampling frequency is 1 Hz or 10 Hz for different occultation cases. In the case of a requirement for a high resolution of the vertical profile, 10 Hz Doppler output will be used, and otherwise, 1 Hz Doppler output will be used. In the two-day experiments in 2022 in this article, 10 Hz sampling was used.

### 4.2. Evaluation of Doppler Observation Accuracy

The Doppler O-C (Observed minus Computed) residuals emerged as the outcome of subtracting the computed theoretical frequency (C) from the observed frequency (O). Within these residuals lay the combined effects of both the Martian and Earth's media influences. The influence exerted by Earth's media can be rectified through the delay correction of the ground station (equipped with water vapor microwave radiometers and GNSS dual-frequency receivers), while the influence attributed to Mars' media acts as the input for subsequent inversion phases. Before evaluating the Doppler O-C residual, we first verified the accuracy of the theoretical value calculation model.

### 4.2.1. Accuracy Verification of Theoretical Values

The calculation of the theoretical value allows the calculation of the theoretical observation frequency introduced by kinematics, which mainly involves the influence of the motion of the spacecraft relative to the observation station (the influence caused by the gravitational redshift and gravitational delay is relatively small and can be ignored here), including the orbital motion of the spacecraft, the orbital motion of the target celestial body

(Mars), the motion of the station relative to the center of the Earth (nutation, precession), and the orbital motion of the Earth. Theoretical received frequency can be expressed as (without gravitational redshift and gravitational delay) [32]:

$$\begin{cases} f_1 = f_0\left(1 - \frac{v}{c}\right) \\ f_{2/3} = f_0\left(1 - \frac{v_1}{c}\right)\left(1 - \frac{v_2}{c}\right) \end{cases} \tag{7}$$

The first formula is the one-way case, and the second one is the two- or three-way case. $f_0$ is the transmit frequency, which is from the spacecraft for the one-way case and from the uplink station for the two-way or three-way case. $v$ is the relative velocity of the downlink station to the spacecraft in the one-way tracking case. $v_1$ is the relative velocity of the spacecraft to the uplink station and $v_2$ is the relative velocity of the downlink station to the spacecraft. The two formulas are simple cases without considering gravitational redshift or gravitational delay influence. In planetary occultation computation, influences that are no greater than 1 mHz can be ignored.

The LEVEL02 data released by Mars Express MEX can be used to verify the calculation accuracy of the Doppler theoretical value. The MEX orbital eccentricity is 0.571, the period is 7.5 h, the periareion altitude is 298 km, and the apoareion altitude is 10,107 km. Here, we used the occultation data of MEX in 2005 (the LEVEL02 format; the 10th column gives the calculated value):

URL (https://pds-geosciences.wustl.edu/mex/mex-m-mrs-1_2_3-v1/mexmrs_0727/data/level02/open_loop/dsn/dpx/m65rsr0l02_dpx_053631939_00.tab, accessed on 26 September 2023)

Corresponding orbit:

URL (https://naif.jpl.nasa.gov/pub/naif/MEX/kernels/spk/ORMM__051201000000_00203.BSP, accessed on 26 September 2023)

And additional spice kernels used in the computation include the time kernel, the Earth orientation kernel, the ground station kernel, and the planetary kernel. All these kernels can be found in the dataset.

The difference between two different theoretically calculated values is given in Figure 6.

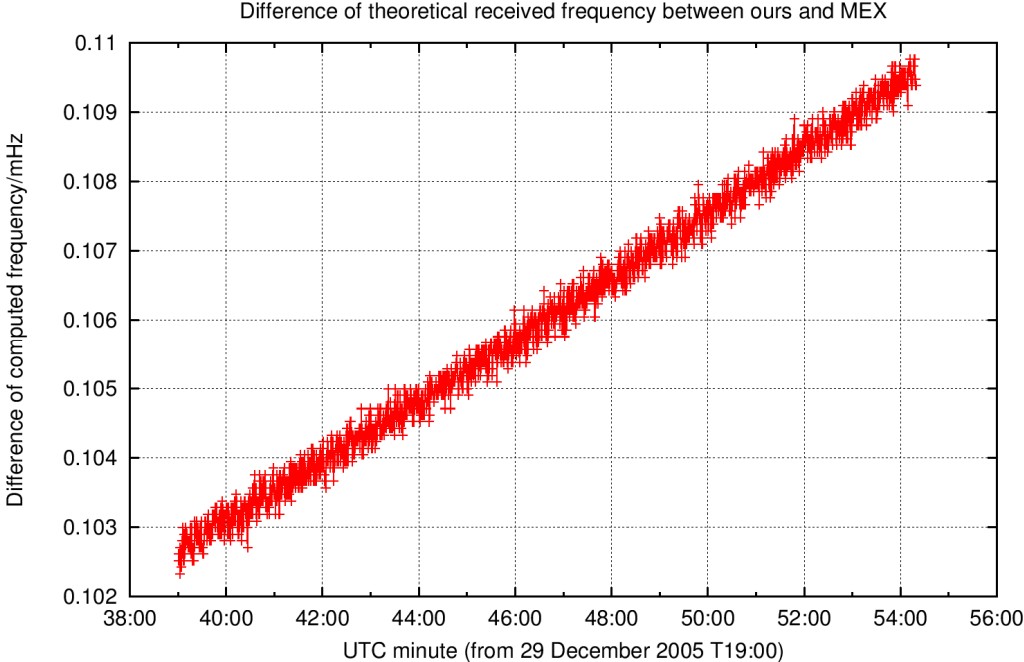

**Figure 6.** With the validation of the Doppler model (two-way observation used), the linear variation could be introduced by the different planetary ephemeris.

In our analysis, we conducted a comparison of the theoretical received frequencies between our approach and the Mars Express (MEX) method under the two-way tracking mode. Notably, the equations for the two-way and three-way modes are equivalent due to their shared Doppler calculation formulas. Given the substantial uncertainty resulting from the random walk noise in one-way tracking data inversion, we focused our evaluation solely on the two-way data, using MEX LEVEL02 data as an external reference standard. The theoretical value calculation of one-way mode is similar to that of two- or three-way mode and can be calculated uniformly through Equation (7). As depicted in Figure 6, the discrepancy between our calculated theoretical value and MEX's theoretical value (LEVEL02) amounts to approximately 0.1 mHz. This linear variation in the figure may arise from differences in the planetary ephemeris utilized. With observation noise standing at approximately 15 mHz, it becomes evident that the accuracy of our theoretical model aligns with the requisites for successful inversion processes.

### 4.2.2. Doppler Residual Evaluation

As the theoretical value was validated, Doppler O-C residuals for all the one-way, two-way, and three-way observations were evaluated as follows.

Figure 7a provides an overview of the Doppler residual distribution concerning the occultation height within arc-1, which employs a one-way tracking approach. Given the relatively modest stability of the onboard USO at $10^{-11}$ (at one second) and $10^{-10}$ (over the span of one day), inherent random walk noise becomes pronounced in the residuals, manifesting at an amplitude of approximately 40 mHz. Due to such noise, the inversion of the atmosphere is not reasonable, so our one-way observation was only given the results for the ionosphere.

In contrast, Figure 7b outlines the distribution of Doppler residual outcomes corresponding to arc-4, conducted under a two-way tracking mode with a residual noise of 15 mHz. Compared with the one-way measurement, the accuracy of the two-way observation showed great improvement. We used the two-way observation data to retrieve the parameters of the Martian ionosphere and neutral atmosphere.

Figure 7c shows the distribution of the three-way Doppler residuals with occultation height in the arc-5. The accuracy of the three-way residual was consistent with that of the two-way, and the noise amplitude was about 15 mHz. As a comparison, we also used the three-way tracking of the data to retrieve the neutral atmosphere and ionospheric parameters of Mars. Notably, the detailed images in Figure 7b,c offer insights into a transition spanning from 110 km to 200 km.

Both arc-4 and arc-5 share congruent tracking time spans, implying a similar impact of the Martian medium on the Doppler signal. This consistency should theoretically extend to both observations. Figure 7d showcases a comparative assessment of the residuals from these two observations. Notably, the amplitude of the observed differences nearly aligns with the amplitude of the observed noise. This correspondence underscores the agreement between the two sets of residuals.

### 4.3. Inversion Results of the Neutral Atmosphere and Ionosphere from Tianwen-1

We developed atmospheric and ionospheric inversion software for Tianwen-1 observations. The inversion results of one-way occultation (ingress) for arc-1 and arc-6 observations, as well as the inversion results of two-way occultation (egress) for arc-4 and arc-9 observations and three-way occultation (egress) for arc-5 and arc-10 observations, are given as follows.

In April 2004, the MEX mission began conventional radio occultation observations. The results of Tianwen-1 observations are compared with those of MEX. The MEX-0112 data are the result of a two-way observation released by ESA. The data for MEX could be searched at URL (https://archives.esac.esa.int/psa, accessed on 26 September 2023).

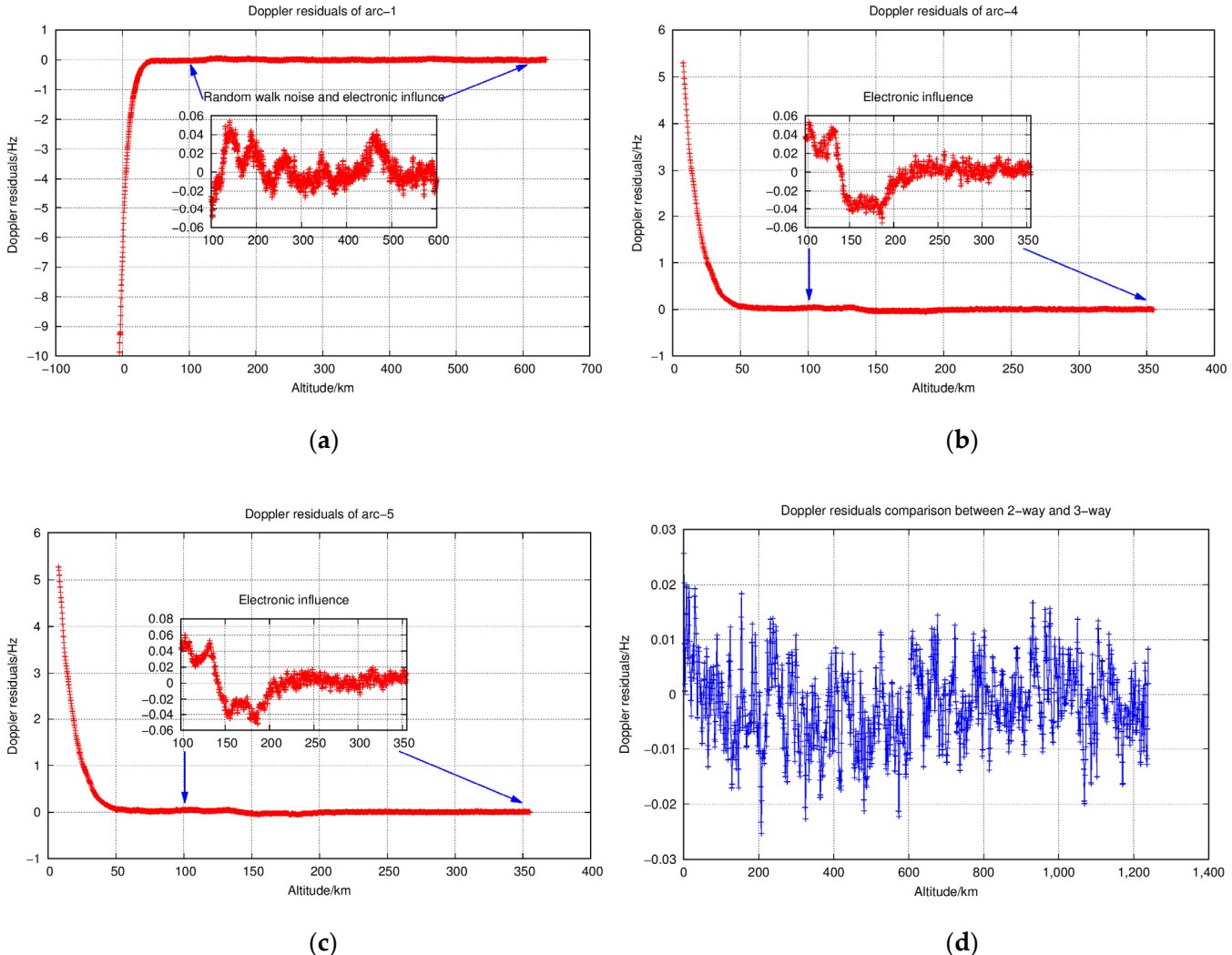

**Figure 7.** Distribution of the Doppler residuals with the occultation height: (**a**) is for the arc-1(ingress) with one-way observation; (**b**) is for the arc-4(egress) with two-way observation; (**c**) is for the arc-5(egress) with three-way observation; and (**d**) is the difference in the arc-4 and arc-5 Doppler residuals. The tracking span is the same for arc-4 and arc-5. Two sessions used two-way and three-way observation modes, respectively, and the Earth media correction was conducted.

4.3.1. Results from One-Way Observation

The arc-1 and arc-6 observations with one-way mode (see Table 1) are used for iono-spheric electron density inversion here.

Based on the Solar Zenith Angle details provided in Table 2, it is evident that all the observations corresponding to arc-1 and arc-6 were conducted during the dayside phase. Typically, dayside electron density profiles of Mars exhibit distinct layers, including the M1 layer below and the M1 layer, M2 layer, and M2 layer above. The primary peak of electron density is situated at approximately 140 km altitude, identified as the M2 layer. This layer is most prominently detected in observations due to its characteristics [33].

Analyzing the electron density profiles in Figure 8, there are peaks below 200 km in arc-1 and arc-6 observations. However, the electronic density is supposed to decrease rapidly above the M2 layer (above 230 km), and a perplexing peak area was observed at an altitude of 450 km in Figure 8. Consequently, the retrieved electron density profiles are not reasonable for either arc-1 or arc-6 observations.

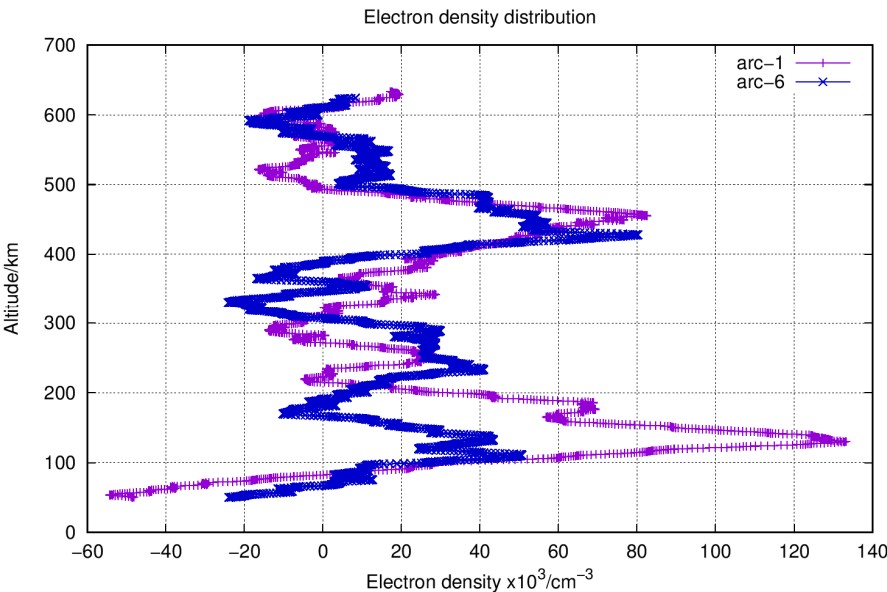

**Figure 8.** Unreliable profiles of the ionospheric electron density with the occultation height for the arc-1 and arc-6 observations with one-way mode.

Upon examination of our theoretical value computation and data processing procedures, we ascertained that no inherent sources could give rise to the observed large-scale periodic structural discrepancies and electron density variations. This investigation has led us to attribute these deviations primarily to the onboard USO, and the details are given as follows.

The carrier measurement residual frequency, the O-C residual frequency, and the electron density with the corresponding occultation altitude on 22 and 25 March are distributed in Figures 9 and 10, respectively.

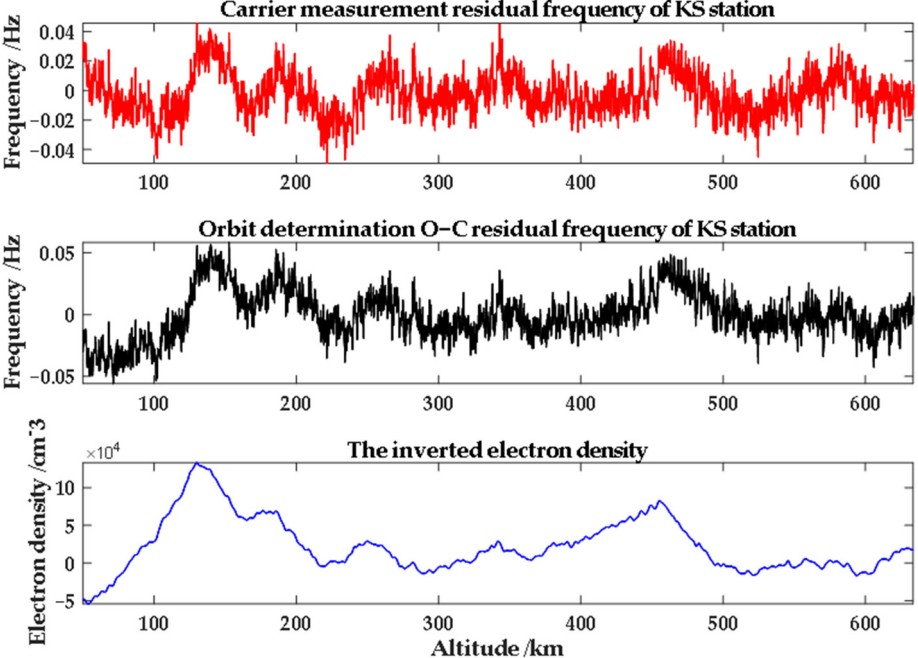

**Figure 9.** The result of comparative analysis of arc-1 on March 22 in 2022: (1) the fitting residual frequency of carrier measurement at KS station; (2) one-way orbit determination of O-C residual frequency at KS station; and (3) the inverted electron density result (x-axis is the altitude of corresponding occultation points).

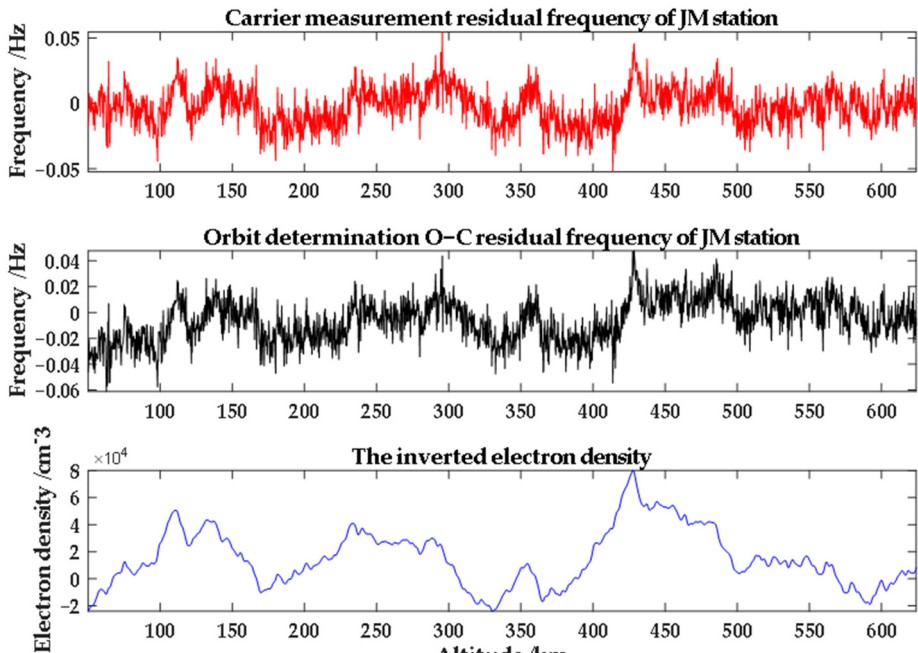

**Figure 10.** The result of comparative analysis of arc-6 on March 25 in 2022: (1) the fitting residual frequency of one-way carrier measurement at JM station; (2) the orbit determination O-C residual frequency at JM station; and (3) the inverted electron density result (x-axis is the altitude of corresponding occultation points).

Analyzing the raw one-way Doppler frequency measurement data, the residual frequencies obtained from fitting clearly showed random irregular fluctuations, which also exist in the orbit residuals (O-C) with a significant effect, leading to irregular fluctuations in the ionospheric electron density of Mars inverted from the O-C residuals [34,35].

From the irregularly fluctuating measurement data, the orbit altitude of Tianwen-1 with the highest electron concentration between 400 and 500 km altitude was about 02:28:01 on 22 March 2022, and the orbit altitude of Tianwen-1 with the highest electron concentration between 400 and 500 km altitude was about 01:16:55 on 25 March 2022. It is inferred that the frequency stability index of the onboard crystal oscillator of Tianwen-1 is not high, and it may cause fluctuations in the one-way transmitted signal, resulting in random irregular fluctuations in the measurement noise in one-way measurements and seriously affecting the reliability of ionospheric electron density inversion through one-way measurements.

It should be noted that prior to obtaining the O-C orbit residuals, the effects of the troposphere and ionosphere on the Earth's stations had already been deduced using actual atmospheric data sampled by stations. And the hardware link characteristics of the stations are also very stable over long periods of time. However, irregular fluctuations still occurred in the O-C orbit residuals that corresponded to those in the raw one-way measurement residuals. Meanwhile, modeling and eliminating these irregular fluctuations in one-way measurement is extremely difficult because the irregular fluctuations are random.

As the fluctuations exist in the residuals of one-way measurements irregularly, we cannot model the inversion results well and fix the abnormal electron density data. In particular, there is a strange signal at an altitude of 400–500 km, which does not conform to the distribution pattern of the Mars ionosphere. It has been confirmed that it is caused by the random walk noise of the onboard USO. We used the data smoothing process for the false structure at 400–500 km. After the data smoothing, the strange signals were removed, and the new distribution of electron density for arc-1 and arc-6 is shown in Figure 11.

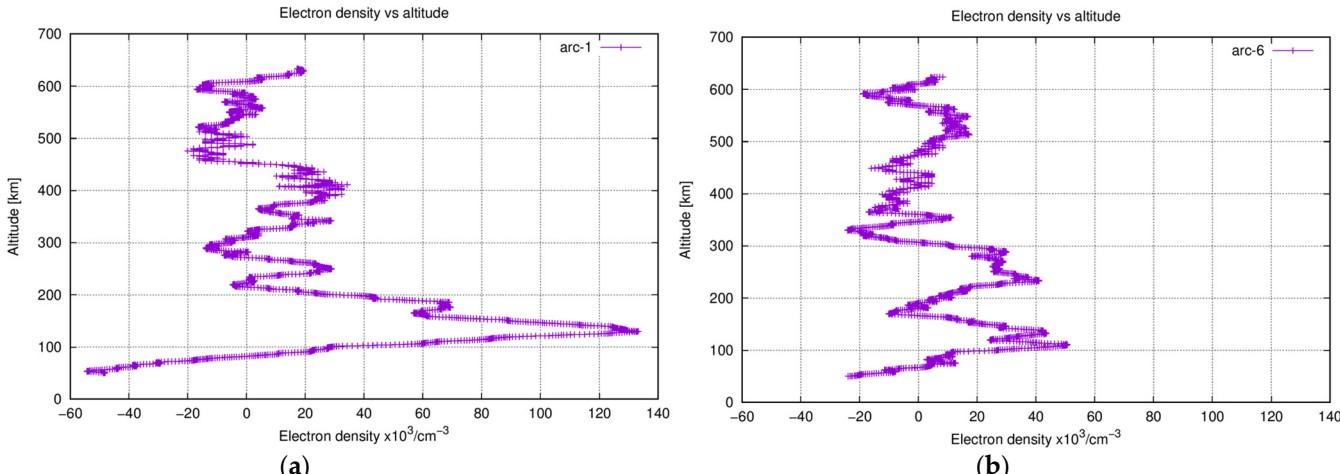

**Figure 11.** Profiles of the ionospheric electron density with the occultation height: (**a**) is arc-1 with one-way observation mode and (**b**) is arc-6 with one-way observation mode.

Due to the random walk noise of USO in one-way mode, the retrieved neutral atmosphere profiles of Mars from Tianwen-1 were also not very reasonable, so we have not presented our inversion results in the text.

Thus, a one-way mode cannot be utilized for effective occultation observations. In contrast, the two-way and three-way modes can avoid USO-related issues, leading to improved observations.

### 4.3.2. Results from Two-Way and Three-Way Observation Modes

Arc-4, arc-9, and MEX-0112 represent instances of the two-way tracking mode, whereas arc-5 and arc-10 embody the three-way tracking mode. Compared with one-way Doppler observations, the two/three-way Doppler residuals are smaller (seen in Figure 7), making the retrieved neutral atmospheric density, temperature, and ionospheric electron density more reliable.

Arc-4, arc-5, arc-9, and arc-10 correspond to egress occultations, while MEX-0112 stems from ingress occultations (as indicated in Table 1). The observations of arcs 4, 5, 9, and 10, being egress occultations, fail to provide profiles of atmospheric parameters below 7 km.

In terms of neutral atmospheric results above 7 km, the displayed features in Figure 12a–c exhibited reasonable consistency with the outcomes of MEX-0112's X-band observations. Both neutral atmospheric density and pressure profiles manifested a logical exponential decrease with increasing altitude. Within our experiment, we assumed the upper boundary temperature to be 165 K, and this assumption is reflected in Figure 12c, wherein the temperature distribution ranges from 7 km to 40 km. However, owing to the absence of observations below 7 km, the planetary boundary layer (below 5 km) remains indistinct in our visualizations.

Taking into account the Solar Zenith Angle information highlighted in Table 2, it is crucial to acknowledge that all observations for arc-4, arc-5, arc-9, arc-10, and MEX-0112 occurred during the dayside. This led to reasonable temperature fluctuations spanning from 165 K to 200.8 K. Beyond 20 km, which corresponds to the ozone layer in Mars' atmosphere, temperature variations become more pronounced due to the influence of solar UV irradiance [36]. These temperature alterations displayed both latitudinal and local variations, with latitude specifics provided in Table 2. Notably, an inversion temperature layer was apparent in the observations of arc-9 and arc-10 at a height of 30 km, which was much higher than that of MEX and arc-4,5. This discrepancy may be attributed to the local time variation of the neutral temperature on Mars, while the SZA differences between MEX and arc-9,10 are greater than those for arc-4,5, as presented in Table 2.

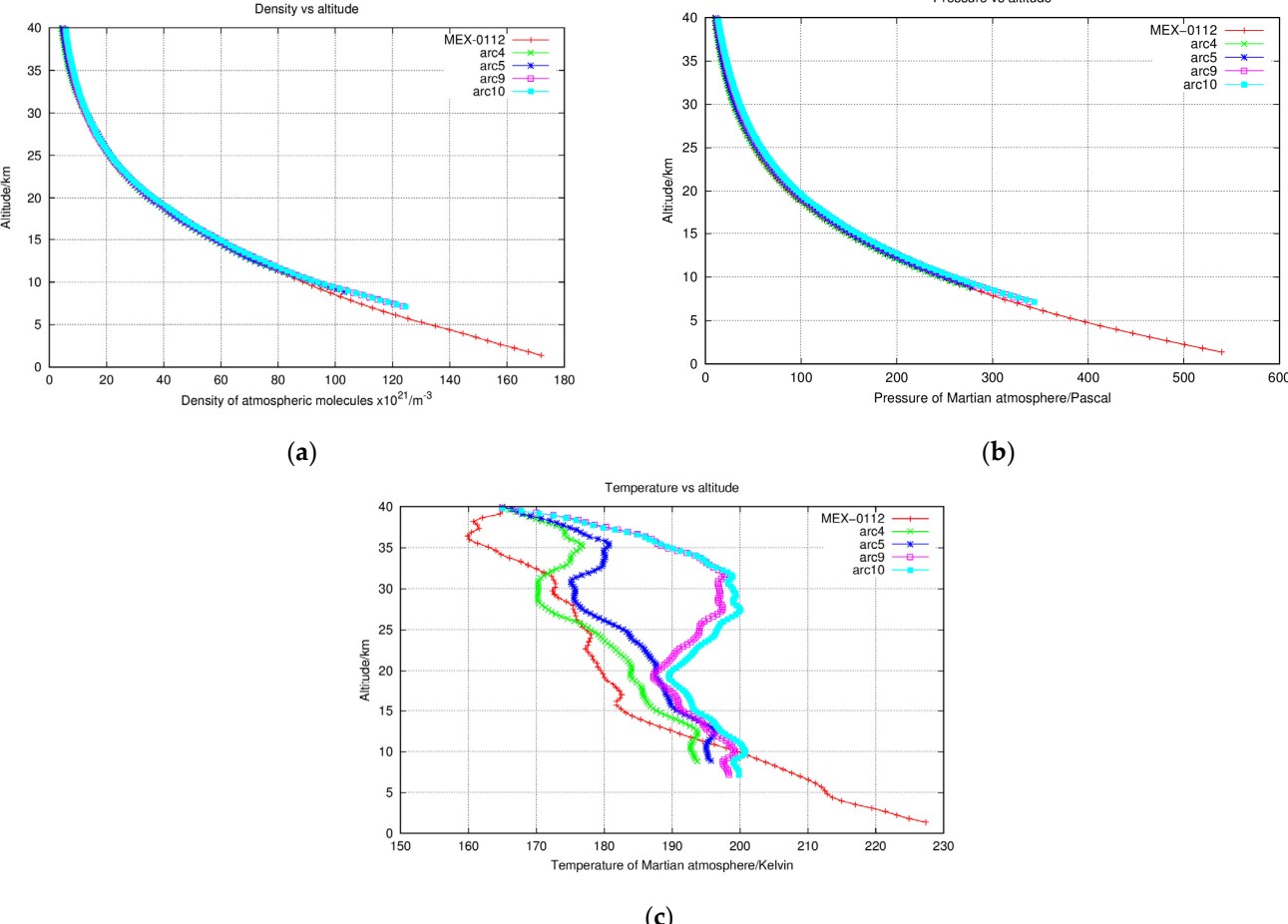

**Figure 12.** (**a**) The atmospheric molecule density profiles of MEX-0112 and arc-4,5,9,10; (**b**) the atmospheric pressure profiles of MEX-0112 and arc-4,5,9,10; and (**c**) the atmospheric temperature profiles of MEX-0112 and arc-4,5,9,10.

Our occultation experiments suffered from data limitations, as depicted in Figure 13. Arc-4 and arc-5 lacked observations from 350 km upward, while arc-9 and arc-10 lacked data from 200 km. This data gap restricts a comprehensive assessment of the upper ionosphere, unlike the broader MEX observations spanning 40 km to 700 km. However, electronic density within the M2 layer was successfully detected in all observations.

The electron density within the M2 layer is notably influenced by factors such as the SZA, solar irradiance, neutral atmospheric densities, etc. And the peak electron density $D_m$ and its altitude $Z_m$ of the M2 layer could be described by Chapman theory [37]:

$$\begin{cases} D_m = D_0 \cos^n(\text{SZA}) \\ Z_m = Z_0 + 10 \times \ln \sec(\text{SZA}) \end{cases} \tag{8}$$

While $D_0 = 2 \times 10^5$ cm$^{-3}$, n = 0.57 and $Z_0 = 120$ km, the coefficient $D_0$, n, $Z_0$ is the simulated results from a previous Mars mission. The mean Solar Zenith Angles were extracted from the mean value from Table 2, playing a vital role in these calculations and contributing to the comprehensive understanding of the electron density distribution in the M2 layer.

However, Equation (8) is the function of the ideal ionosphere with a horizontally stratified atmosphere. This function is valid for small SZAs. For large SZA, the curvature of the planetary atmosphere becomes increasingly important in the ionospheric formation process, especially for SZA larger than $70°$ [38].

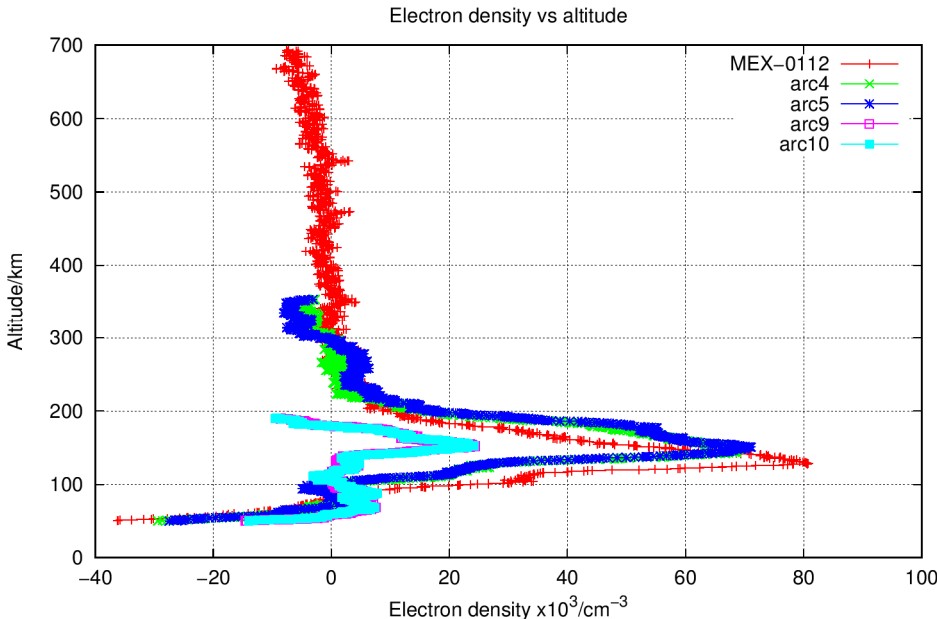

**Figure 13.** The electron density profiles of MEX-0112 and arc-4,5,9,10.

Therefore, the expression $\sec(\text{SZA})$ in Equation (8) should be replaced by corrected functions $\text{ch}(X, \text{SZA})$, when $70° \leq \text{SZA} \leq 90°$ for arc-4,5,9,10 and MEX-0112 observations [39]. The corrected function is given as follows (when $\text{SZA} \leq 90°$):

$$\text{ch}(X, \text{SZA}) = \left(\frac{\pi}{2}X\right)^{1/2} \frac{1.0606963 + 0.55643831y}{1.0619896 + 1.7245609y + y2} \tag{9}$$

where $X = r/H$, and $r$ is the radial distance to the observation point; $H$ is the scale height. $y = \left(\frac{1}{2}X\right)^{1/2}|\cos(\text{SZA})|$ and $0 \leq y < 8$.

More details about Equation (9) can be found in [39]. For $\text{SZA} > 90°$ or $8 \leq y \leq 100$, the expressions of $\text{ch}(X, \text{SZA})$ can also be found in [39].

Equation (8) from the Chapman theory and corrected function Equation (9) was employed to assess inversion results in arc-4,5,9,10, and MEX-0112 observations. These evaluations are detailed in Table 5. This strategy helps us understand ionospheric behavior in these altitude ranges.

**Table 5.** The theoretical and observed peak electronic density and its altitude in the M2 layer of arc-4,5,9,10, and MEX-0112 observations.

| Arc No. | Mean SZA (Degree) | Theoretical Density (cm$^{-3}$) and Altitude | Observed Density (cm$^{-3}$) and Altitude | Density Difference |
|---------|-------------------|----------------------------------------------|-------------------------------------------|---------------------|
| arc-4 | 80.7 | 70,770/138 km | 69,272/148 km | 2% |
| arc-5 | 80.7 | 70,770/138 km | 71,100/150 km | 1% |
| arc-9 | 88.7 | 23,111/150 km | 24,388/152 km | 5% |
| arc-10 | 88.7 | 23,111/150 km | 24,219/152 km | 5% |
| MEX-0112 | 78.0 | 81,670/136 km | 80,791/128 km | 1% |

All the electronic density profiles in Figure 13 illustrate the distinct M1 and M2 layers, showing the electronic density's rapid decrease beyond the M2 layer. With altitude as the variable, the electronic density fluctuations in arc-4,5 and arc-9,10 exhibited consistency with the observations made by MEX-0112. This indicates that our observations align with established trends in the Martian ionosphere.

In Table 5, the theoretical peak electronic density and corresponding altitudes within the M2 layer for arc-4,5 and arc-9,10 are outlined and complemented by results from MEX-0112. The differences between theoretical and observed peak electronic densities remained within the range of 1200 cm$^{-3}$ (less than 5%). Furthermore, the variations in altitude discrepancies were confined to less than 12 km. The accuracy of our observations in capturing key ionospheric characteristics could be proven.

The electronic density profiles derived from the observations in arc-4,5, and arcs-9,10 revealed a logical distribution across altitudes, closely adhering to Chapman theory's expectations for peak M2 layer values. These findings collectively validate the efficacy of the two-way and three-way modes to avoid the USO-related issue. More significantly, the comparable precision of both modes underscores their potential to enhance the accuracy of observations, thereby contributing to our deeper understanding of the Martian ionosphere.

## 5. Conclusions

Our study has achieved success through the acquisition of radio occultation observations from Tianwen-1 over two distinct days. The validation process of our Doppler computation model, achieved through comparison with MEX-released data, has further fortified the robustness of our approach. Utilizing the observation data derived from one-way, two-way, and three-way tracking modes, we embarked on the task of inverting the profiles of both the Martian neutral atmosphere and ionosphere. Our analysis reveals that, due to the inherent random walk noise intrinsic to the onboard USO, one-way tracking data regrettably failed to yield valuable results. In contrast, the comprehensive utilization of two-way and three-way tracking data yielded favorable outcomes, demonstrating coherence between the two methodologies.

From altitudes of 0 km to 40 km, we inverted the neutral atmospheric profiles. However, compared with the inversion results of MEX's ingress occultation, our inversion results of Tianwen-1 egress observations did not extend to altitudes below 7 km. The atmospheric parameters were reasonably distributed above 7 km, which closely aligns with MEX's findings in the X band.

Above the altitude of 40 km, we inverted the ionospheric electron density profiles. The occultation observations of Tianwen-1 revealed a peak region of electron density at an altitude of about 140 km, a result that concurs with MEX's findings. The peak electron densities in the M2 layer of different observations and their altitudes were accessed using Chapman theory, and they exhibited strong variations, potentially attributed to diurnal shifts in the Mars ionosphere.

Furthermore, there was an inversion temperature layer above the 20 km observed from our experiment. However, it is impossible to provide evidence of cross-validation due to the lack of corresponding information. Therefore, further confirmation of the signal requires subsequent observations. We hope to promote more joint observations involving Tianwen-1, aiming to generate more inversion outcomes of the Martian ionosphere and neutral atmosphere, thereby facilitating comprehensive analyses of Martian atmospheric characteristics.

In summary, this research not only showed the viability of the radio occultation methodology in the context of the Tianwen-1 mission but also emphasized the importance of multi-faceted tracking modes for robust results. By achieving agreement between the two-way and three-way tracking modes, our study adds an essential contribution to the knowledge surrounding Martian atmospheric and ionospheric profiles.

**Author Contributions:** Conceptualization, M.L. and L.C.; methodology, N.J., J.P. and L.C.; software, N.J., P.G. and Q.H.; validation, L.C., N.J. and J.K.; formal analysis, J.P. and P.G.; investigation, L.C., N.J. and M.W. (Mengjie Wu); resources, L.C. and M.W. (Mei Wang); data curation, M.W. (Mei Wang), J.K. and L.C.; writing—original draft preparation, M.L. and N.J.; writing—review and editing, P.G. and L.C.; visualization, N.J.; supervision, L.C.; project administration, M.L.; funding acquisition, M.L. All authors have read and agreed to the published version of this manuscript.

**Funding:** This research was funded by the National Natural Science Foundation of China (42005099, 11973015, 11833001) and the National Key R&D Program of China (grant number 2020YFA0713501).

**Data Availability Statement:** The data presented in this study are available on request from the corresponding author.

**Acknowledgments:** The authors appreciate NASA and ESA for the provision of relevant data and products.

**Conflicts of Interest:** The authors declare no conflict of interest.

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
