# Peer review of "Preliminary Estimations of Mars Atmospheric and Ionospheric Profiles from Tianwen-1 Radio Occultation One-Way, Two-Way, and Three-Way Observations"

_remotesensing, doi:10.3390/rs15235506_

Round 1

Reviewer 1 Report (Previous Reviewer 1)

Comments and Suggestions for Authors

The revised manu could be accepted. 

I know it is the first time for China to make the Martian radio occultation measurements. But the data quality is a little bit lower than thoes obtained by the RO experiments on MAVEN, MEX. For example, the profiles arc-4,5 shown in Figure 12 are limited within 360km. How about the profiles above 360km, which are very important to determine the accuracy of the RO data.

Comments on the Quality of English Language

Though the grammar mistakes are corrected. It should be noted that there are still a lot of Chinese English in the revised manu.

Author Response

Reviewer 2 Report (New Reviewer)

Comments and Suggestions for Authors

1.The detailed orbit determination strategy and parameter settings should be given.

2.Besides evaluating the accuracy of orbit determination through 0-C, can you use the method of orbit overlap to verify the accuracy of orbit determination.

Author Response

Reviewer 3 Report (New Reviewer)

Comments and Suggestions for Authors

The paper is interesting to show both operational and technical capabilities of Tianwen-1 to perform Radio science experiments. The data taken at 2 days are processed to test the procedure and get doppler residual which  are inversed to get the atmospheric profile . The discussion  of  individual results (i.e 2 profiles ) is important for validation, but the steps to obtain them must be explained in detail. The are  information missing in the  the  description of the experiment (including the HW, upping downlink frequencies, etc etc )  and the validation of data processing ( media corrections, choices of BC for inversion). It is nevertheless good news hat 2-way  RO has the potential but the profiles  missing uncertainty analysis.  I liked in particular  to use of MEX data for the validation to conversion to doppler residual but I would like to see more details how the authors go from LV0 to -LV2..   the MEX data can be used for LV2 validation but   is not ideal for higher level i.e to compare with the Electron density or lower atm results since the season, location and local time are different.  The authors can find my detailed comments and suggestions in the attached pdf. 

Comments on the Quality of English Language

There are only minor issues...

Author Response

This manuscript is a resubmission of an earlier submission. The following is a list of the peer review reports and author responses from that submission.

Round 1

Reviewer 1 Report

Comments and Suggestions for Authors

Two big problems:

(1) the quality of the English writing of the manuscript.  In my last review, I just said that: "The English writing of the revised manuscript is still far from acceptable. For the sake of saving time, I just give some examples as follows". The authors just corrected the examples I listed. They are just examples of the errors I found in the first two page of the manuscript. Actually, there are a lot of error and Chinese English expressions in the rest of the manu. Please find a person with good english writing skills or a company to polish or rewrite the manuscript before the resubmission.

(2) the problem of Equation (8). The authors added the corrected function (Equation 9) and the description. However, the theoretical peak electron densities and altitudes listed in Table 3 in the revised manuscript are the same as those in the old manuscript. The SZAs of arc-4/5/9/10 are larger than 80° and even close to 90°. It is unbelievable that the peak electron densities and altitudes be the same with the corrected functions.

Please solve these two problems before making the resubmission.

Best

Comments on the Quality of English Language

Please see the words above.
